# A Missense Variant in *PDK1* Associated with Severe Neurodevelopmental Delay and Epilepsy

**DOI:** 10.3390/biomedicines10123171

**Published:** 2022-12-07

**Authors:** Raquel Vaz, Josephine Wincent, Najla Elfissi, Kristina Rosengren Forsblad, Maria Pettersson, Karin Naess, Anna Wedell, Anna Wredenberg, Anna Lindstrand, Sofia Ygberg

**Affiliations:** 1Department of Molecular Medicine and Surgery, Karolinska Institute, 17177 Stockholm, Sweden; 2Department of Clinical Genetics, Karolinska University Hospital, 17177 Stockholm, Sweden; 3Department of Medical Biochemistry and Biophysics, Karolinska Institute, 17177 Stockholm, Sweden; 4Akademiska Sjukhuset, 75185 Uppsala, Sweden; 5Centre for Inherited Metabolic Diseases, Karolinska University Hospital, 17177 Stockholm, Sweden

**Keywords:** pyruvate dehydrogenase kinase, epilepsy, neurodevelopmental delay, zebrafish

## Abstract

The pyruvate dehydrogenase complex (PDC) is responsible for the conversion of pyruvate into acetyl-CoA, which is used for energy conversion in cells. PDC activity is regulated by phosphorylation via kinases and phosphatases (PDK/PDP). Variants in all subunits of the PDC and in PDK3 have been reported, with varying phenotypes including lactic acidosis, neurodevelopmental delay, peripheral neuropathy, or seizures. Here, we report a de novo heterozygous missense variant in *PDK1* (c.1139G > A; p.G380D) in a girl with developmental delay and early onset severe epilepsy. To investigate the role of PDK1^G380D^ in energy metabolism and neuronal development, we used a zebrafish model. In zebrafish embryos we show a reduced number of cells with mitochondria with membrane potential, reduced movements, and a delay in neuronal development. Furthermore, we observe a reduction in the phosphorylation of PDH-E1α by PDK^G380D^, which suggests a disruption in the regulation of PDC activity. Finally, in patient fibroblasts, a mild reduction in the ratio of phosphorylated PDH over total PDH-E1α was detected. In summary, our findings support the notion that this aberrant PDK1 activity is the cause of clinical symptoms in the patient.

## 1. Introduction

The pyruvate dehydrogenase complex (PDC), located in the mitochondrial matrix, is responsible for the irreversible conversion of pyruvate into acetyl-CoA upon glycolysis, fueling the tricarboxylic acid (TCA) cycle. The PDC is composed of three enzymes: pyruvate dehydrogenase or PDH (E1), dihydrolipoyl transacetylase (E2), and FAD-containing dihydrolipoamide dehydrogenase (E3) [1], and its activity is tightly regulated to reassure the energy needs of the cells are met. The key enzyme at the center of PDC activity control is PDH, whose activity is controlled by the balance between the activity of PDH kinases and phosphatases, that in turn sense the metabolic state of the cell [2,3]. PDH is a tetramer composed of 2a and 2b subunits [4], in which the a subunits are the target for phosphorylation at Ser232, Ser293, and Ser300 [5,6]. Tissue-specific pyruvate dehydrogenase kinases (PDKs) phosphorylate PDH-E1α subunits and render PDC inactive. Conversely, pyruvate dehydrogenase phosphatases (PDPs) revert that inactivation by removing the phosphate groups [7]. Four PDKs (PDK1-4) have been found in humans and, according to expression analysis in humans and rodents, they are found at highest levels in the cardiac and skeletal muscle. PDK1, PDK2, and PDK4 are also found in other organs such as the brain, pancreas, lung, or kidneys [8,9]. In the skeletal muscle, the most abundant isoforms are PDK2 and PDK4; in the brain, northern blot analysis showed that PDK2 is the most abundant isoform, followed by PDK1. Conversely, the in vitro kinetic analysis of PDKs activity showed that the isoforms with the highest kinase activity are either PDK1 or PDK3, depending on the publication, followed by PDK4 and lastly PDK2 [9,10]. Interestingly, the rate of reactivation of PDC by PDPs is dependent on the PDK that phosphorylated PDH, with the fastest dephosphorylation occurring following PDK2 activity, then PDK3, PDK4, and finally PDK1 [10]. The activity of the kinases depends on the ratio of CoA/acetyl-CoA, NAD/NADH, as well as the amount of pyruvate present, with high levels of acetyl-CoA and NAHD stimulating the activity of PDK and, consequently, inactivating PDC [2,11,12]. Furthermore, the extracellular environment also affects PDC regulation, as *PDK1* transcription is directly upregulated by HIF-1 (hypoxia-inducible factor-1) during hypoxia, to favor anaerobic catabolism [13,14].

Previous publications have reported that patients with dysregulated pyruvate dehydrogenase activity typically present with neurological features, such as microcephaly, developmental delay, seizures, and ataxia, that appear during early childhood. Such patients are less well suited to use sugar as a source of energy, as pyruvate is not converted to acetyl-CoA, causing an energy deficiency. Instead, pyruvate is converted to lactate by lactate dehydrogenase [15]. As neurons are a major energy consuming cell type, they are severely affected by a reduction in PDC activity [16]. To date, most of the variants affecting the PDC have been found in *PDHA1* (PDH-E1α), with fewer variants found in *DLAT* (E2) and *PDHX* (E3) [17,18,19,20,21,22,23,24,25,26,27]. In two unrelated families, a hemizygous missense variant in *PDK3* was associated with Charcot-Marie-Tooth disease, affecting the peripheral nervous system [28,29]. The mechanism involved hyperphosphorylation and concomitant reduced PDC activity, which impaired mitochondrial axonal transport. Furthermore, two brothers have been reported to each carry a homozygous 3-nucleotide deletion variant in *PDP1* and another patient to carry a homozygous 1-nucleotide duplication with a clinical phenotype including lactic acidosis, neonatal hypotonia, and feeding difficulties [30,31,32]. Current therapies of PDC deficiency involve the administration of co-factors such as thiamine to optimize residual enzyme activity and implementing a ketogenic diet, bypassing the need for PDC activity to produce energy [26,33,34,35].

In this study, we present a girl with dysmorphic facial features, severe developmental delay, and seizures. Using trio whole genome sequencing (WGS), we found a de novo missense variant in *PDK1*, suggested to be the cause of the disease. The expression of the variant in zebrafish leads to a reduced number of cells with detectable mitochondrial membrane potential, a reduced muscle activity, and overall defective neurodevelopment. The findings support a role for mutant PDK1 in this patient and suggest *PDK1* as a novel disease-causing gene.

## 2. Materials and Methods

### 2.1. Clinical Investigations

The patient was followed up at Uppsala University Hospital. The parents provided informed consent for the child’s participation in this study, which has been approved by the local ethical board in Stockholm, Sweden (ethics permit number KS 2012/222-31/3).

### 2.2. Whole Genome Sequencing and Sanger Sequencing

Whole genome sequencing of the proband and both parents was performed using a 30× PCR-free paired-end WGS protocol on an Illumina HiSeq 2500 platform as described previously [36]. In brief, the variants were prioritized based on conservation, frequency in internal and public databases, and inheritance. The ranked variants were then visualized in the Scout analysis platform [37].

The *PDK1* variant was confirmed by PCR and Sanger sequencing using the forward primer 5′ GGTTATGGATTGCCCATATCAC 3′ and reverse primer 5′ TTCATTACAGTTAAGTATGG 3′. The Sanger sequencing was performed on an ABI 3730 PRISM^®^ DNA Analyzer.

### 2.3. Cell Culture and Oxygen Consumption Analysis

A skin biopsy was obtained from which fibroblasts were grown. The PDH activity of the unstimulated cells was measured in the lab of Gary Brown, Oxford Regional Genetics Laboratories, Oxford University Hospital NHS Foundation Trust using a locally available analysis, available upon request [38]. After the maximal activation of PDC by dichloroacetate, the cells are incubated with [1-^14^C] pyruvate and the resulting ^14^CO_2_ generated is quantified in a liquid scintillation counter. The normal range for this assay is 0.6–0.9 nmol/mg protein/minute.

### 2.4. Zebrafish Studies

#### 2.4.1. Zebrafish Husbandry

The wild type adult zebrafish were maintained at the Zebrafish Core facility (KM-C, Karolinska Institutet) according to standard procedures and with permission from the Stockholm Ethical Board for Animal Experiments (ethics permit number 13063-2017 and 14049-2019).

#### 2.4.2. Pdk1 Knockdown and PDK1 mRNA Synthesis

The endogenous *pdk1* expression was knocked down using a translation blocking morpholino (5′ AAGTCCTGAAGATCCTCATGTTGGC 3′) [39]. For complementation studies, human wild type (*wt*) and mutant *PDK1* (c.1139G > A, p.G380D) mRNA were used. The coding sequences for both *wt* and mutant were synthesized in vitro and cloned into the pCDNA 3.1(+) plasmid using the EcoRI and XhoI cloning sites (Biomatik, Kitchener ON, Canada). For in vitro mRNA transcription, the plasmids were first linearized by digestion with XhoI, followed by transcription using the T7 mMessage machine kit (AM1344, Invitrogen, Thermo Fisher Scientific, Waltham, MA, USA).

#### 2.4.3. Embryo Collection and Injection

The embryos were produced by light-induced mass spawning, collected, and injected at the 1–2-cell stage. The embryos were injected with 0.5 mM *pdk1*-morpholino (*pdk1*-MO) and 0.5 mM *pdk1*-MO combined with 200 mg/mL *wt* or *p.G380D PDK1* (*pdk1*-MO + *PDK1^wt^* or *pdk1*-MO + *PDK1^G380D^*, respectively).

The control and injected embryos were maintained in an incubator at 28.5 °C. From one day post fertilization, the embryo water was supplemented with 30 mg/L PTU (1-phenyl 2-thiourea; P7629, Merk, Sigma, St. Louis, MO, USA). PTU inhibits the development of the melanocytes [40,41], maintaining the transparency of the embryos and facilitating imaging. The bright field images of larvae at 3 days post fertilization were taken using an Evos LX core microscope with an in-built camera.

#### 2.4.4. Mitochondria Labeling

The mitochondria were labeled with the membrane potential sensitive dye Mitotracker (M22426, Invitrogen). The dechorionated one-day-old embryos were exposed to the Mitotracker diluted in embryo water supplemented with 30 mg/L PTU at a final concentration of 1 mM for 24 h at 28.5 °C protected from light.

After exposure, the embryos were washed several times with embryo water, followed by tissue fixation using 4% paraformaldehyde (PFA; HL96753.1000, HistoLab, Gothenburg, Sweden) overnight at 4 °C. The embryos were then washed in 1× phosphate-buffered saline (P4417, Merk) supplemented with 0.1% Tween-20 (P1379, Merk) or PBST and subsequently imaged using a Zeiss LSM710 confocal. The quantification was performed by counting the number of Mitotracker-positive cells in an area that extends 300 μm starting at the pronephric duct towards the tail of the embryo.

#### 2.4.5. Spontaneous Movement Analysis

One-day-old embryos were used to assess spontaneous unstimulated movement, or coiling. From ten to fifteen chorionated embryos were placed in a Petri dish with embryo water and recorded for 5 min, using a Leica epifluorescence microscope (DMC2900, Wetzlar, Germany) coupled with a Leica DFC450C camera. The number of coils per 5 min for each embryo and experimental group was quantified and compared.

#### 2.4.6. Inhibition of PDK Function

PS10 (2((2,4-dihydroxyphenyl)sulfonyl)isoindoline-4,6-diol, HY-121744, MedChemExpress, Monmouth Junction, NJ, USA) was used to block PDK activity [42,43]. The compound was reconstituted in DMSO (dimethyl sulfoxide, D8418, Sigma) and injected into 1–2-cell stage embryos at a concentration of 1 mM, as determined by a previous dose/toxicity experiment.

#### 2.4.7. Electron Microscopy

Conventional transmission electron microscopy was performed on three-days-old zebrafish embryos as previously published [44]. The sections from the trunk of the embryos, that contained mostly skeletal muscle, were imaged and abnormalities in skeletal muscle and mitochondria were assessed.

#### 2.4.8. Immunohistochemistry (IHC) and Imaging

The embryos at the desired developmental stage were fixed with 4% PFA overnight at 4 °C. The embryos were subsequently dehydrated in an increasing series of methanol until transferred to 100% methanol and stored at −20 °C. Following rehydration, IHC protocol was followed as previously described [45]. The primary antibodies used were anti-acetylated tubulin (T6793, Sigma) at 1:400 and anti-myosin (A4.1025, deposited to the DSHB by Blau, H.M., University of Yowa, Iowa City, IA, USA) at 1:50, and the secondary antibodies used were a goat anti-mouse Alexa488 (A11029, Invitrogen, Waltham, MA, USA) at 1:200 in blocking solution. For nuclear staining, DAPI (4′,6-diamidino-2-phenylindole; D1306, Invitrogen) at 30 mg/mL was added to the secondary antibody’s solution.

The embryos were mounted in a drop of 1% low melting agarose (16520-050, Invitrogen) in 1 × PBS for correct embryo orientation and were imaged using a water dipping lens in a Zeiss LSM710 confocal microscope.

### 2.5. Western Blot

Zebrafish protein was extracted from a pool of 15–20 de-yolked embryos and human protein was obtained from a confluent T75 flask of cultured fibroblasts, corresponding to approximately 10^6^ cells. The protein extraction, quantification, and Western blot was performed as previously described [46]. Ten micrograms of total protein were used. The total protein in the blot was quantified using stain-free gels and low-fluorescence PVDF membranes (BioRad, Hercules, CA, USA). Here, the trihalo compound in the stain-free gel binds the tryptophane in the proteins. Following activation with UV light, the total protein present in the gels and in the membranes becomes visible and the membranes are imaged for subsequent quantification. The antibodies used were rabbit anti-PDH-E1α pSer232 (AP1036, Merk), mouse anti-PDHA1 (45-6600, Invitrogen), anti-active caspase3 (559565, BD Biosciences, Stockholm, Sweden), goat anti-rabbit-HRP (A16110, Invitrogen), and goat anti-mouse-HRP (G-21040, Invitrogen). The signal detection was achieved using the Clarity of Clarity Max ECL Western Substrate (170-5060 and 170-562, BioRad) and the BioRad ChemiDoc MP imaging system.

### 2.6. Statistical Analysis

The experiments were performed in triplicate and quantifications were statistically compared using GraphPad Prism version 9. For the data that did not fit the normal distribution of values using the D’Agostino and Pearsons test, a Ln(value) function was applied and the normality was re-tested and confirmed. The statistical analysis of the data that did not follow the normal distribution of values was performed using the Kruskal–Wallis test, followed by the Dunn’s multiple comparison test (statistical analysis for total body length comparison). Conversely, one-way ANOVA was performed when the normality was verified and the Tukey’s multiple comparisons test was performed to identify which groups were significantly different. The statistical significance was considered when *p* < 0.05. Non-significant comparisons were not plotted in the graphs. For each comparison, n represents the number of embryos analyzed per group.

## 3. Results

### 3.1. Clinical Description

The female proband was the first child to healthy, non-consanguineous parents. In retrospect, the mother recalled frequent jerks or hiccups from the fetus from early pregnancy. The routine ultrasound during pregnancy was normal. She was delivered vaginally with vacuum extraction at gestation week 41 + 1, with a birth weight and head circumference at -1SD and length at +1SD. Due to irritability, hypertonic posture, and tachypnea she was admitted to the NICU at one day of age for suspected infection. She recovered and the blood-samples did not indicate an ongoing infection. Shortly after, at two months of age, she developed feeding difficulties and frequent myoclonic jerks. The EEG initially showed a burst-suppression pattern, evolving into hypsarrhythmia already at 2 months of age. She was also noted to have a special appearance with long fingers, a prominent nasal tip, narrow face, and low-set ears. She developed a therapy resistant epilepsy within her first two months of life, with multiple seizure types including myoclonia, absences, and tonic seizures. She had a continuously pathological EEG, however, not filling the LGS EEG criteria. Her development was very slow, she was continuously irritable, and her body tone altered between severely hypertonic and hypotonic. She was on ketogenic diet from age 6 months and for the rest of her life. Concomitant with the introduction of the diet, the number of seizures were reduced. When she was about 2 years old, she was started on Q10 orally and thereafter became relatively seizure free. However, she continued to be severely affected throughout her life and did not reach any developmental milestones beyond the level of 4–6 weeks. Her head growth haltered around 7 months. MRI of the brain at 10 days, 6 months, and at 3 years of age had a normal appearance, although the latter showed some minor dilatation of her ventricles compared to the earlier examinations. At 2 years and 8 months, she was treated in the PICU for a severe pneumonia with multiorgan failure. At 3 years of age, she developed pubic hair and sweat with adult odor, although the basic endocrine laboratory tests were normal. She passed away from a second septic episode during the evaluation of this suspected precocious puberty. Extensive metabolic screening, including clinical trio WGS and analysis with an epilepsy gene panel did not reveal any cause. Muscle biopsy indicated normal ATP-production and normal function of the complexes in the respiratory chain. Muscle histology was normal in clinical routine histological stainings, however, during the autopsy, skeletal muscle investigations by electron microscopy showed the presence of electron dense inclusions (1–2 μm in size) of unknown origin.

### 3.2. De Novo Missense Variant in PDK1

The research analysis of the entire WGS data identified a heterozygous de novo missense variant (c.1139G > A, p.Gly380Asp (OMIM 602524, NM_002610)) in *pyruvate dehydrogenase kinase 1* (*PDK1*) on chromosome 2q31.1. The variant had not previously been reported in GnomAD [47], ExAC [48], or 1000G [49] and had a combined annotation-dependent depletion (CADD) score of 33. The affected amino acid was conserved across species according to PHAST, GERP, and phyloP. The missense substitution was predicted to be deleterious by SIFT and probably damaging by Polyphen. The combined rank score by the Karolinska Clinical Genome Analysis software was 26, indicating a high likelihood of clinical relevance [37].

The Sanger sequencing of the region that was flanking the variant was performed and it confirmed the presence of the heterozygous variant in the patient but not in the parents (Figure 1A).

### 3.3. Lower Enzymatic Activity of PDH in Cultured Patient Fibroblasts

The activity of PDH in patient fibroblasts was measured in an enzymatic assay that revealed a slightly lower enzymatic activity 0.58 nmol/mg protein/min (ref 0.60–0.90 nmol/mg protein/min). The conditions were standard and did not involve stressing the cells.

### 3.4. PDK1^G380D^ Has No Relevant Effect on the Growth of Zebrafish Embryos

To understand how the expression of *PDK1^G380D^* affects development, the zebrafish model was used. Zebrafish have one orthologue for *PDK1* sharing over 77% similarity at the amino acid level with the human protein and high conservation surrounding the amino acid affected by the variant (Figure 1B). The similarity in the protein sequence suggests that the function is also conserved between species. Rescue experiments in zebrafish embryos were performed by overexpressing wild type or mutant human *PDK1* (*PDK1^wt^* and *PDK1^G380D^*) in embryos with reduced endogenous Pdk1 by using a translation blocking morpholino (*pdk1*-MO) [39]. Firstly, we investigated whether the presence of PDK1^G380D^ would affect the early stages of embryonic development. The injection of *pdk1*-MO + *PDK1^G380D^* did not result in any visible defects in embryo development when compared to the other experimental groups (control, *pdk1*-MO and *pdk1*-MO + *PDK1^wt^*, Figure 2). Even though the larvae in the experimental groups were statistically smaller than those in the control group, the larger difference between control and *pdk1*-MO + *PDK1^wt^* is 161 μm, which corresponds to a 5% difference in body length. This finding, combined with a normal appearance of the larvae, suggests that the expression of mutant *PDK1* does not have a major impact in the overall embryo growth during early development.

### 3.5. Mutant PDK1 Affects the Membrane Potential of Mitochondria and Muscle Function

PDK1 phosphorylates the E1a component of PDH, a key component in providing the substrates necessary for the TCA cycle in the mitochondria. In order to investigate the mitochondrial network in the presence of PDK1^G380D^, we labeled the zebrafish embryos with a Mitotracker labelling assay. Briefly, viable mitochondria tightly regulate their membrane potential within physiological levels, which is important for ATP production and storage and, consequently, cell viability. Mitotracker, a membrane potential-dependent mitochondria dye, was therefore used to quantify the metabolically active mitochondria. Surprisingly, the quantification of positively labeled cells revealed a lower number of cells containing mitochondria with membrane potential in embryos injected with *pdk1*-MO and *pdk1*-MO + *PDK1^G380D^* when compared to the other experimental conditions for the same area in the tail of two-days-old embryos (area analyzed: 300 × 350 µm width × height, *p* < 0.0001) (Figure 3A, square). The ultrastructure of the mitochondria was also compared using electron microscopy and we found no difference in the total number nor the structure of the mitochondria between the experimental groups (Appendix A).

The membrane potential in the mitochondrion is vital for the homeostasis of this organelle [50]. Therefore, the reduced number of cells with labeled mitochondria seen in the embryos expressing *PDK1^G380D^* suggests that the energy conversion may be affected. Since the energy supply is essential for skeletal muscle function, we assessed muscle function in zebrafish embryos expressing mutant *PDK1*. At early developmental stages, typically 1 day post fertilization, the zebrafish embryos present with spontaneous muscle activity driven by the spinal cord and independent of signals originating from the brain [51]. Spontaneous activity was recorded (Appendix A), quantified, and revealed that embryos co-injected with *pdk1*-MO + *PDK1^G380D^* mRNA move significantly less times for the duration of the recording (5 min) when compared with the other experimental groups (*p* < 0.0001, Figure 3B). Importantly, the reduced spontaneous activity was not due to defects in the skeletal muscle, as the tissue structure was intact in the embryos from all the experimental groups (Appendix A). These finding indicate that the loss of PDK1 results in fewer mitotracker positive cells and that this phenotype can be rescued by wildtype PDK1 albeit not mutated PDK1. Additionally, in the embryos injected with *pdk1*-MO + *PDK1^G380D^*, we detected reduced coiling as compared to both morpholino and wildtype.

### 3.6. PDK1 Is Important for Neuronal Differentiation Progression

The patient described here presented with severe symptoms affecting the central nervous system. To assess if a similar presentation is seen in zebrafish, we immunolabeled the neurons of embryos at early developmental stages (three days post fertilization) and found that the number of neuronal connections in the tectum was lower for the *pdk1*-MO + *PDK1^G380D^* group than in the other experimental groups (*p* < 0.05, Figure 3C). A reduction in the number of neuronal connections can be due to developmental delay or degeneration of the neurons. To assess this, we quantified the apoptosis in the different experimental groups and found that the increase in apoptosis in the experimental groups compared to the control did not fully explain the phenotype (Figure 3D). This increase may, to an extent, be a result of some toxicity due to the injection of the morpholino and mRNAs in the embryos and not necessarily their role in apoptosis. The findings suggest that the reduction in the number of neuronal connections at the developmental stage analyzed is mostly due to a delayed neuronal development, which may include a failure in axonal outgrowth.

### 3.7. Variant in PDK1 Impairs Phosphorylation of PDH-E1α

PDK1 phosphorylates the E1a subunit of PDH. To further investigate the suspected hypomorphic nature of PDK1^G380D^, we used PS10, a compound that blocks phosphorylation of PDH-E1α [42,43], in combination with *pdk1*-MO and the human mRNA. The quantification of the number of cells containing mitochondria with membrane potential in the injected embryos showed that PDK^wt^ did not rescue the phenotype in the presence of PS10, as expected, while the number of cells with labeled mitochondria was unchanged when the PS10 was co-injected with either *pdk1*-MO or *pdk1*-MO + *PDK^G380D^* (*p* < 0.0001, Figure 3E). To confirm the reduction in phosphorylation by mutant PDK1 in the patient, we quantified the amount of phosphorylated PDH, PDH-E1α pSer232, in fibroblasts using Western blot (Figure 4). When compared to control samples, we found a reduction in the ratio of phosphorylated protein over total PDH-E1α. These results support the hypothesis that PDK1^G380D^ is less able to phosphorylate PDH-E1α.

## 4. Discussion

We describe a patient with severe epilepsy where genetic analysis showed a de novo missense variant in *PDK1*, a gene not previously associated with disease. To investigate the role of PDK1^G380D^ in energy metabolism and neuronal development, we used a zebrafish model. In zebrafish embryos injected with *pdk1*-MO + *PDK1^G380D^*, we found a reduction in the number of cells containing mitochondria with membrane potential, a reduced spontaneous movement, and affected neuronal development. Furthermore, our results suggest that the disease mechanism involved the reduction in the ability for PDK^G380D^ to phosphorylate PDH-E1α, which we suggest affects the regulation of PDC activity and, therefore, the homeostasis of energy conversion in the mitochondria.

Variants in several genes involved in PDC activity have been found to cause disease. Most reported patients carry variants in *PDHA1* [19,26,34,35,52,53,54,55,56], followed by *DLAT (E2)* [17,21], *PDHX (E3)* [18,23,25,27], and *PDK3* [28,29,31] and to a smaller extent *PDH-E1b* [20] and *PDP1* [22,30,32]. Here, we report for the first time a disease-causing variant in *PDK1*. Despite variants in the six genes coding for structural components of the PDC or enzymes that regulate PDH activity, the symptoms overlap to a great extent. In all the cases, a decrease in PDH activity is detected, including the variant in *PDK3*, which was reported as a gain-of-function variant [29]. It is, therefore, surprising that our patient presented with a de novo suspected hypomorphic variant in *PDK1*, as seen by a reduction in the phosphorylation of PDH-E1α in patient cells, but having a similar clinical presentation as the patients with a reduced PDC activity. In parallel, the studies have shown an increased expression of PDK1 in cancer cells, which deactivates the PDC by phosphorylation. This deactivation is suggested to result in a change in the cells’ energy conversion toward aerobic glycolysis instead of mitochondrial respiration, also known as the Warburg effect [57,58,59]. PDK1 is therefore a good target for cancer therapies, for which inhibitors have been tested. Several compounds able to inhibit PDK1 function resulted in either proliferation inhibition or apoptosis of cancer cells [6,60,61,62]. Even though we did not detect a significant increase in apoptosis in zebrafish embryos when expressing *PDK1^G380D^* compared to the other groups, we suggest that cells in both the patient and zebrafish embryos may be similarly affected by the lower phosphorylation of PDH. A decrease in the number of mitochondria with membrane potential in the zebrafish embryos in the presence of PDK1^G380D^ suggests some level of regulation of mitochondrial function or cell viability. However, when the electron micrographs were analyzed, we did not find a significant difference in the number of mitochondria nor the presence of structural defects in this organelle. Several publications have reported on the absence of mitochondrial phenotypes, such as the number, localization, size, or structure [22,52], which is consistent with our results but that we do not fully understand. In embryos where the Pdk1 is reduced or when mutant *PDK1* is expressed, we found a decrease in the spontaneous muscle activity, or coiling. Given that no structural defects were found in the skeletal muscle, we suggest that there may be a defect in energy conversion, likely from a decrease in functional mitochondria. As glycolysis results in less energy conversion than the mitochondria-driven OXPHOS [63], which, we suggest, accounts for the reduction in spontaneous skeletal muscle contraction.

Altogether, the compiled results suggest that the clinical phenotype of the patient as well as the phenotypes observed in the zebrafish are mediated by abnormalities in the PDC activity. However, since the available antibodies to detect the phosphorylated PDH do not work in zebrafish, we were not able to confirm this mechanism. This limitation needs to be taken into account. Another major issue is the variability of phenotypes observed in both morphant and *PDK1^wt^* overexpressing embryos. It is unlikely that the transient approach used in this study will be able to address such questions. Further experiments using patient cells and stable mutant animal models would be better suited to fully explain the mechanisms of PDK1-associated disorders.

The patients with variants in genes coding for components involved in PDC activity present with neurological symptoms, including Leigh syndrome, neurodevelopmental delay, ADHD, and brain malformations as shown by MRI. Nevertheless, the effect of such variants in neurodevelopment remains poorly understood. Previous publications have shown that PDH is highly expressed in neurons but not in astrocytes, which express preferentially lactate dehydrogenase, involved in an alternative energy conversion pathway than OXPHOS [64]. Another particularity in the brain is that the TCA cycle also produces glutamate and g-aminobutyric acid (GABA), which are important neurotransmitters for synapsis [65]. A detailed characterization of a brain-specific *Pdha1* knockout mouse model showed a decrease in glutamate in the brain and decreased electrical activity, suggesting that the TCA activity is reduced [66]. A study in rats showed that glutamatergic neurons consume more energy than GABAergic neurons [67], which suggests that glutamatergic neurons are especially susceptible to the reduced TCA activity resulting from the low PDC activity. Moreover, disrupted glutamatergic activity and increased aerobic glycolysis have been associated with epilepsy [68,69], a common clinical presentation in patients, including the girl described in this work. We suggest, therefore, that the presence of PDK1^G380D^ in the brain may affect neuronal differentiation and activity by disrupting energy and neurotransmitter synthesis.

The current treatment options are limited and used to a large degree indiscriminate of the genetic diagnosis. These include the introduction to a ketogenic diet often coupled with the supplementation of thiamine, carnitine, and dichloroacetic acid (DCA) [16,30,33,34,54]. For a number of patients, the treatment is efficacious in reducing lactic acidosis and the number of seizures. However, this is not true for all patients, which either do not respond to the treatment, or it fails due to the normal progression of the disease or when patients develop other illnesses, such as infections [26,52], also observed in our patient. It is, therefore, essential to develop better therapies that not only are more efficacious but also target the specific pathology of each patient. A zebrafish knockout for *dlat*, which codes for the PDH-E2 protein, recapitulated the patients’ clinical presentations, such as increased lactate and pyruvate. Feeding the larvae with a ketogenic diet resulted in a rescue of the PDH-deficiency including improved swimming behavior, similar to wild type larvae [70], which strongly supports the use of this model to test new candidate therapies.

## 5. Conclusions

Here, we report for the first time a disorder caused by a variant in *PDK1*, which was generally characterized by neurodevelopmental delay and epilepsy. Modeling the disorder in zebrafish embryos, we were able to describe in more detail the phenotypes resulting from the expression of mutant *PDK1* in vivo, which is not possible when using, e.g., patient fibroblasts. The development of such models will facilitate the testing of more efficacious targeted therapies for patients, taking into consideration the genetic diagnosis instead of treating the clinical presentations as they are identified.

## Figures and Tables

**Figure 1 biomedicines-10-03171-f001:**
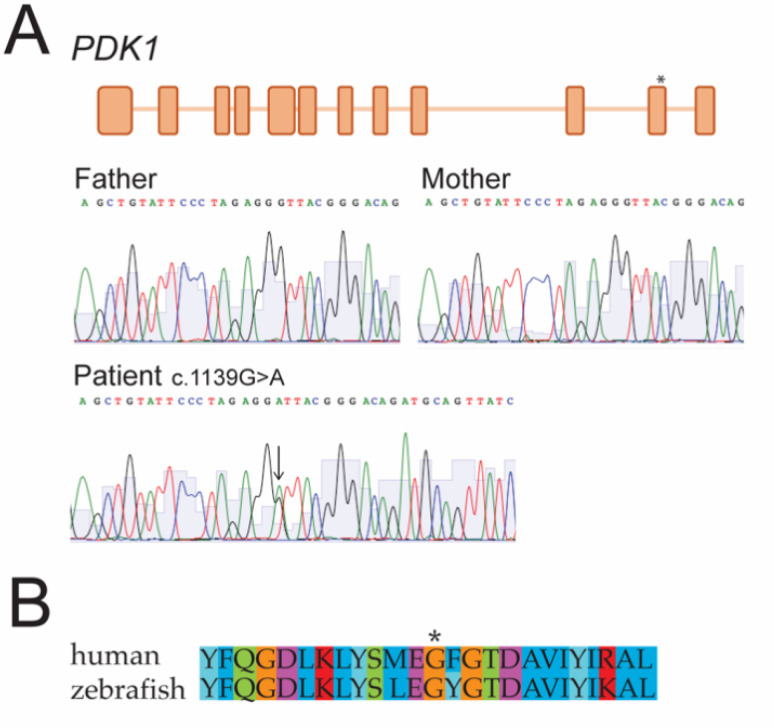
(**A**) Genome analysis revealed a de novo variant in *PDK1*. Whole genome sequencing showed a c.1139G > A (p.G380D) variant in exon 11 of *PDK1* (*) in heterozygosity (arrow). Sanger sequencing of the parents and the proband confirmed it was a de novo variant. (**B**) Partial amino acid sequence alignment including the amino acid affected by the variant (*) shows conservation between human and zebrafish.

**Figure 2 biomedicines-10-03171-f002:**
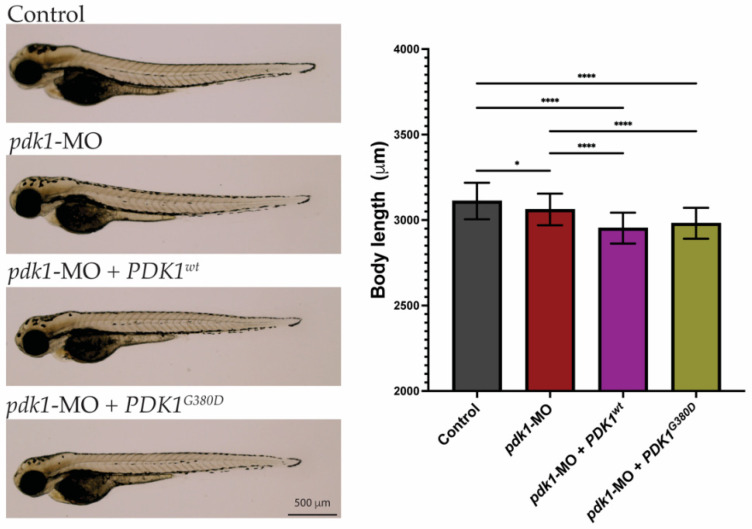
Morphology of zebrafish embryos. Three-days-old embryos from all experimental groups were photographed and show no major phenotypes. Quantification of total body length showed embryos from the experimental groups were significantly smaller than those from the control group (n = 50–75 embryos per group, * *p* < 0.026, **** *p* < 0.0001).

**Figure 3 biomedicines-10-03171-f003:**
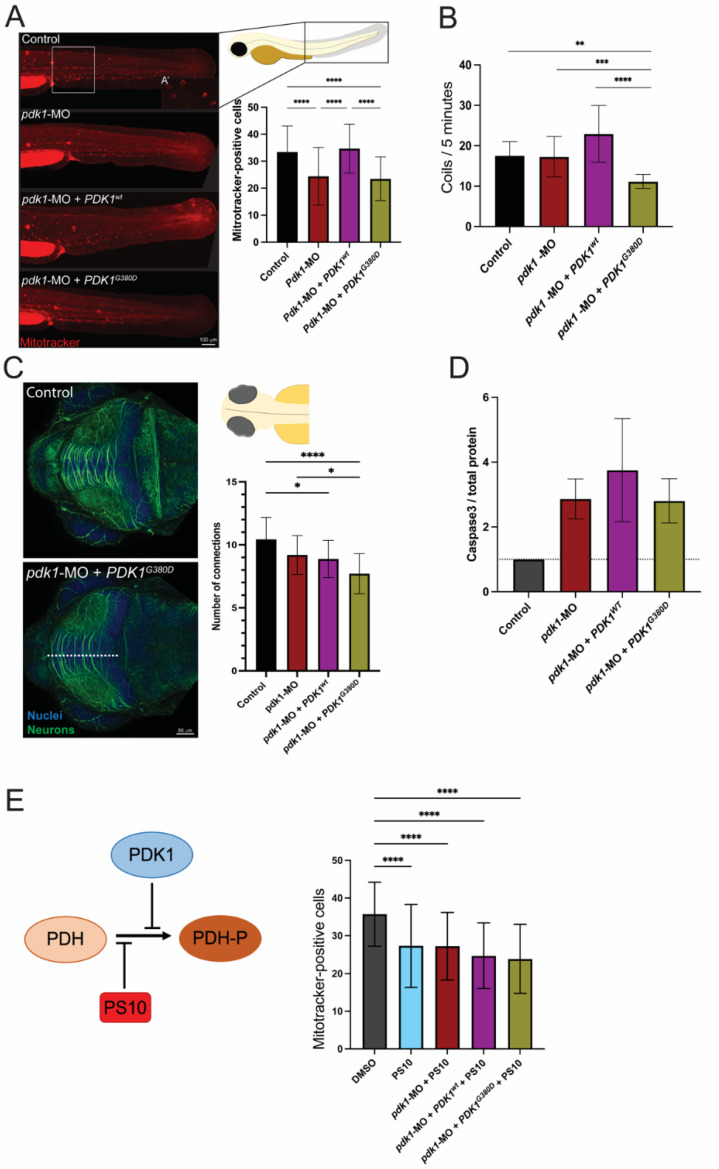
Expression of *PDK1^G380D^* in zebrafish embryos affects embryogenesis. (**A**) Mitochondria with membrane potential were labeled using a Mitotracker labeling assay and the quantification was performed by counting the positively stained cells (see inset for close up on positive cells) in an area of 300 × 350 µm in the trunk of the embryo (white box). (A’) Mitotracker-positive cells shown in higher magnification. A significant decrease in the number of cells with labeled mitochondria was found in *pdk1*-MO + *PDK1^G380D^* when compared to control and *pdk1*-MO + *PDK1^wt^* embryos (n = 40–45 per group, **** *p* < 0.0001). (**B**) Expression of mutant *PDK1* in zebrafish embryos resulted in a significant decrease in spontaneous movements (coils) when compared to the others experimental groups (n = 108–120 embryos per group, ** *p* = 0.003, *** *p* = 0.004, **** *p* < 0.0001). (**C**) Neuronal differentiation was assessed by labelling embryos with acetylated-a-tubulin (neurons, in green) followed by quantification of the neuronal projections in the tectum (top view). We found a significant decrease in the number of projections crossing the white dotted line of embryos injected with *pdk1*-MO + *PDK1^G380D^* when compared to control embryos (n = 30 embryos per group, * p_control vs. *pdk1*-MO_ = 0.011 and p_control vs. *pdk1*-MO + *PDK1wt*_ = 0.0214, **** *p* < 0.0001). (**D**) Apoptosis was increased in experimental groups when compared to control embryos, detected by Western blot, but it does not correlate with the severity of the other phenotypes analyzed (quantification of cleaved Caspase-3 normalized to total protein, n = 4 protein extracts per group). (**E**) PS10, a compound known to inhibit the phosphorylation of PDH-E1α upstream of PDKs, inhibits the rescue by PDK^wt^, and supports the hypomorphic nature of PDK1^G380D^ (n = 52–70 embryos per group, **** *p* < 0.0001).

**Figure 4 biomedicines-10-03171-f004:**
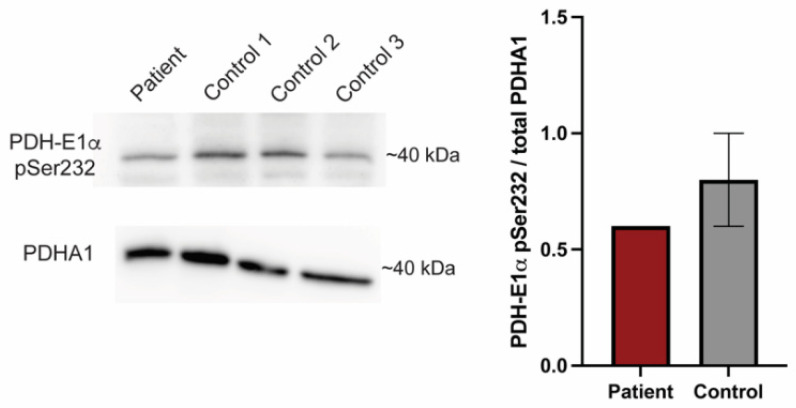
PDK1^G380D^ causes a reduction in phosphorylated PDH-E1α. Phosphorylated PDH-E1α (PDH-E1α pSer232) is reduced in protein extracts from our patient fibroblasts when compared to three controls, which is consistent with the findings in zebrafish. The quantification was normalized to total PDHA1.

## Data Availability

The ethical approval did not permit the sharing of GS data. Access to deidentified data not provided may be requested via the corresponding author.

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
