# Peer review of "A Missense Variant in PDK1 Associated with Severe Neurodevelopmental Delay and Epilepsy"

_biomedicines, 2022, doi:10.3390/biomedicines10123171_

Round 1
Reviewer 1 Report
In the paper “A missense variant in PDK1 associated with severe neurodevelopmental delay and epilepsy” Vaz et al describe a child with a de novo, never before reported, variant in the pyruvate dehydrogenase kinase enzyme 1 (PDK1). PDK1 phosphorylates serines in pyruvate dehydrogenase (PDH) and by doing so inactivate PDH. The child developed a phenotype reminiscent of children with pyruvate dehydrogenase complex (PDC) deficiency. She responded moderately well to the ketogenic diet as would be expected for children with PDC deficiency. They then engage in a logical series of experiments using the zebra fish embryo model to show that the
I greatly enjoyed reading this paper. The paper is well written, and the logical experimental design is masterfully executed. Because the paper is so rich in data, the approach I took to reviewing it was a stepwise descriptive approach, jotting down my questions as I progressed in the paper. I realize that this will make this review somewhat long. To make this read a bit easier I have bolded all my questions.
The authors begin with a clinical history.
Question regarding the clinical history: The authors state that “shortly after” the child developed hypsarrhythmia. I am curious to know at what age she did so. Indeed, hypsarrhythmia is diagnostic for epileptic spasms and it would be highly unusual for a child to develop epileptic spams before 2 months of age. If she did, it should be reported.
Question regarding the clinical history: The child developed multiple types of seizures. This is highly reminiscent of Lennox-Gastaut syndrome. Did the EEG meet criteria for LGS?
They then show that the child had a de novo heterozygous variant in the PKD1 gene.
Question regarding the missense variant: It would be interesting to know what the prediction of the sequencing analysis software was. The glycine to aspartate variant is a non-conservative variant with a small hydrophobic amino acid substituted for a hydrophilic acidic amino acid. Was this predicted to be a pathogenic variant?
Cultured fibroblasts from a skin biopsy showed slightly reduced PDH enzymatic activity.
Question regarding the PDH enzymatic activity: The decrease in PKD1 activity would be expected to result in decreased PDH phosphorylation. Since PDH phosphorylation inactivates the enzyme, this result is somewhat counterintuitive. How do the authors explain this finding? Also, is this very slight decrease in enzyme activity clinically meaningful?
The authors then use translation blocking morpholino to suppress endogenous PKD1 activity in zebra fish and then attempt to rescue the phenotype by overexpressing the wild type PKD1 or mutant PKD1 carrying the same variant as the child. Interestingly, expressing the mutant PKD1 has no effect on embryo growth or ultrastructure of the mitochondria. However, when they labelled mitochondria in the same preparation, they were able to show that the embryos with the mutant rescue had fewer mitochondria and decreased spontaneous movements.
Question regarding the mitochondria labeling: The authors mention that the assay quantifies mitochondria with membrane potentials. To an uninitiated reader, such as me, this is difficult to follow. Is there such a thing as a mitochondrion with no membrane potential? Or is this a descriptive characteristic of the assay that it targets membrane potential. It would be helpful to have a very short sentence clarifying this.
Suggestion regarding figure 2A: The legend states that quantification was done in the trunk of the embryo and refers to the box in the figure. Furthermore, the figure refers the reader to an inset for a close up on positive cells. It is not clear which box the legend refers to and I cannot find the close up of the insets. It might be useful to eliminate the two insets if they are not being used and to leave the box in the schematic.
Question regarding figure 2B: The control and PKD1 morpholino have the same amount of spontaneous movement, while adding the PKD1 wild type results in increase in movement and PKD1 G380D decreased movement. It is a bit counterintuitive that the morpholino variant would have as much spontaneous movement as the control. How is this explained?
Question regarding figure 2C: The same type of question arises here. While the morpholino shows fewer neurons crossing the midline, co-expression of the wild type does not appear to rescue the phenotype. Why is this?
Question regarding figure 2D: Apoptosis appears to be increased in all experimental groups. Why is this?
Question: In paragraph 3.6, last line, the authors conclude that “This suggests that the reduction in the number of neuronal connections at the developmental stage analyzed is mostly due to a failure to develop.” Is it not possible that the same result could be due to a failure of axonal outgrowth? Or is this what is meant by “failure to develop”?
The authors then go on to show that the mechanism underlying the deficit in the PKD1 G380D is a failure to phosphorylate PDH. To do this they use a PKD1 upstream blocker of phosphorylation and show that in the zebra fish embryo phosphorylation is blocked regardless of the PKD1 (mutant or wild type) expressed.
Question: The authors state that wild type PKD1 expression in this preparation does not rescue the phosphorylation. I am a bit puzzled by this statement, is this not exactly what would be expected since PS10 blocks phosphorylation upstream of PKD1?
They then show that in a preparation of skin fibroblasts from the patient the phosphorylation level of PDH is decreased.
The discussion is rich and acknowledges that the mechanisms underlying the pathogenicity of variants in the PDC is still poorly understood. Nevertheless, this paper is the first description of a PKD1 pathogenic variant resulting in a PDC deficiency with a phenotype very similar to other patients with PDC disorders. Furthermore, they show very convincingly in a zebra fish model that the defect in PDC is the direct result of failure to phosphorylate PDH and that this failure leads to a defect in neuronal growth.
Author Response
The authors begin with a clinical history.
- Question regarding the clinical history: The authors state that “shortly after” the child developed hypsarrhythmia. I am curious to know at what age she did so. Indeed, hypsarrhythmia is diagnostic for epileptic spasms and it would be highly unusual for a child to develop epileptic spams before 2 months of age. If she did, it should be reported.
We have added the information in the following line:
Line 233: “Shortly after, at two months of age, …”
- Question regarding the clinical history: The child developed multiple types of seizures. This is highly reminiscent of Lennox-Gastaut syndrome. Did the EEG meet criteria for LGS?
We have revisited the EG data and found that she did not fulfill the EEG criteria for LGS.
We added the following sentence in the manuscript:
Lines 239-240: “She had a continuously pathological EEG however not filling the LGS EEG criteria.”
They then show that the child had a de novo heterozygous variant in the PKD1 gene.
- Question regarding the missense variant: It would be interesting to know what the prediction of the sequencing analysis software was. The glycine to aspartate variant is a non-conservative variant with a small hydrophobic amino acid substituted for a hydrophilic acidic amino acid. Was this predicted to be a pathogenic variant?
The information regarding the pathogenicity prediction was added to the manuscript.
Lines 264-268: “The affected amino acid was conserved across species according to PHAST, GERP, and phyloP. The missense substitution was predicted to be deleterious by SIFT and probably damaging by Polyphen. The combined rank score by the Karolinska Clinical Genome Analysis software was 26, indicating high likelihood of clinical relevance [37].”
Cultured fibroblasts from a skin biopsy showed slightly reduced PDH enzymatic activity.
- Question regarding the PDH enzymatic activity: The decrease in PKD1 activity would be expected to result in decreased PDH phosphorylation. Since PDH phosphorylation inactivates the enzyme, this result is somewhat counterintuitive. How do the authors explain this finding? Also, is this very slight decrease in enzyme activity clinically meaningful?
We understand the reviewer’s points. Regarding the first comment, we believe that it may not be that simple. Even if the PDK1 variant will lead to activation of PDC, such overactivation may in context of the cell in the end lead to reduced PDC activity. We have altered the text to more clearly describe our findings, and not claim to have outlined the mechanism of such altered regulation. Regarding the second question. This assay was done using unstimulated fibroblasts, which may be the reason we didn’t see larger differences.
The authors then use translation blocking morpholino to suppress endogenous PKD1 activity in zebra fish and then attempt to rescue the phenotype by overexpressing the wild type PKD1 or mutant PKD1 carrying the same variant as the child. Interestingly, expressing the mutant PKD1 has no effect on embryo growth or ultrastructure of the mitochondria. However, when they labelled mitochondria in the same preparation, they were able to show that the embryos with the mutant rescue had fewer mitochondria and decreased spontaneous movements.
- Question regarding the mitochondria labeling: The authors mention that the assay quantifies mitochondria with membrane potentials. To an uninitiated reader, such as me, this is difficult to follow. Is there such a thing as a mitochondrion with no membrane potential? Or is this a descriptive characteristic of the assay that it targets membrane potential. It would be helpful to have a very short sentence clarifying this.
We appreciate and understand the reviewer’s comment. The understanding in the literature is that the membrane potential can transiently change, however a sustained decrease will affect the organelle homeostasis and consequently the cell viability. There are several dyes available in the market to label mitochondria, both dependent, independent, or partially dependent on the membrane potential.
We chose the Mitotracker dye for two main reasons: 1) we were particularly interested in detecting viable healthy mitochondria and 2) imaging of large specimen as zebrafish embryos is more complex than a layer of cells in a culture dish, therefore it would be beneficial to find a method that would facilitate the imaging and following quantifications.
We have added the following sentences in the manuscript:
Lines 339-343: “Briefly, viable mitochondria tightly regulate their membrane potential within physio-logical levels which is important for ATP production and storage and, consequently, cell viability. Mitotracker, a membrane potential-dependent mitochondria dye, was therefore used to quantify metabolically active mitochondria.”
- Suggestion regarding figure 2A: The legend states that quantification was done in the trunk of the embryo and refers to the box in the figure. Furthermore, the figure refers the reader to an inset for a close up on positive cells. It is not clear which box the legend refers to and I cannot find the close up of the insets. It might be useful to eliminate the two insets if they are not being used and to leave the box in the schematic.
The figure and figure legend were indeed confusing, and we have modified them so it becomes clearer to the reader. We added the higher magnification image so the readers can see in more detail what the staining looks like, especially for those familiar with mitochondrial stainings.
Note that figure 2 is now figure 3.
- Question regarding figure 2B: The control and PKD1 morpholino have the same amount of spontaneous movement, while adding the PKD1 wild type results in increase in movement and PKD1 G380D decreased movement. It is a bit counterintuitive that the morpholino variant would have as much spontaneous movement as the control. How is this explained?
Since the morpholino used in this project had already been tested and published (Sips et al., 2018), and the results demonstrated in Figure 1A showed an effect when compared to the control embryos, we did not further study the differences between control and pdk1-MO, for example, to what concerns the coiling experiments. The morpholino was used to reduce the amount of endogenous Pdk1 to better recapitulate the heterozygous expression of mutant PDK1 in the patient. However, we could expect that injecting a higher concentration of morpholino would result in a difference in the other essays as well. The main focus of these experiments was to perform the rescue with the wild type or the mutant human mRNA, we therefore were more interested in the differences between the pdk1-MO + PDK1wt and pdk1-MO + PDK1G380D.
- Question regarding figure 2C: The same type of question arises here. While the morpholino shows fewer neurons crossing the midline, co-expression of the wild type does not appear to rescue the phenotype. Why is this?
This is a very interesting point. We agree with the reviewer that this particular finding would warrant further investigation. However, in order to study such complex anatomy a stable line is needed which is beyond the scope of this publication. Here, we were focused on comparing the wide type and mutant mRNA.
- Question regarding figure 2D: Apoptosis appears to be increased in all experimental groups. Why is this?
When we planned the experiments, especially the rescue, we used the lower effective concentration of human RNA to inject the embryos. Nevertheless, we suspect that the fact that we are injecting RNA and morpholino in the embryos may result in some toxicity.
Importantly, as mentioned in the first section of the results, we do not see any gross developmental phenotype. Further investigations using a stable knockout/knockin lines would clarify such phenotypes.
We added the following sentence in the manuscript:
Lines 423-425: “… did not fully explain the phenotype (Figure 3D). This increase may, to an extent, be result to some toxicity due to injection of the morpholino and mRNAs in the embryos and not necessarily their role in apoptosis.”
- Question: In paragraph 3.6, last line, the authors conclude that “This suggests that the reduction in the number of neuronal connections at the developmental stage analyzed is mostly due to a failure to develop.” Is it not possible that the same result could be due to a failure of axonal outgrowth? Or is this what is meant by “failure to develop”?
We agree with the reviewer. In fact, we our general term of failure to develop includes all aspects of neuron formation and maturation. We have changed the manuscript as follows:
Lines 427-428: “… due to a delayed neuronal development, which may include a failure in axonal outgrowth.”
The authors then go on to show that the mechanism underlying the deficit in the PKD1 G380D is a failure to phosphorylate PDH. To do this they use a PKD1 upstream blocker of phosphorylation and show that in the zebra fish embryo phosphorylation is blocked regardless of the PKD1 (mutant or wild type) expressed.
- Question: The authors state that wild type PKD1 expression in this preparation does not rescue the phosphorylation. I am a bit puzzled by this statement, is this not exactly what would be expected since PS10 blocks phosphorylation upstream of PKD1?
Exactly. We did decide that it was important to emphasize that finding to show the readers. To clarify we have added “…as expected…” to the original sentence.
Line 436: “… as expected, …”
Reviewer 2 Report
In this manuscript Vaz et al. report a new disease-causing variant in the PDK1 gene identified in a young female patient. In addition, the authors used a zebrafish model to evaluate potential pathological mechanisms underlying the neurological symptoms in the described patient. The authors claim that differences in mitochondrial numbers as well as neuronal development may explain the phenotype of their patient. In general I enjoyed reading and reviewing this manuscript and I think the description of a new disease-causing variant of an important gene such as PDK1 as well as data on pathophysiological effects of this variant are novel and significant findings and are of high interest to the readership of "Biomedicines". However, I do have several concerns regarding the used methodology and conclusions drawn by their data, as well as some suggestions to the authors, which I will explain in the following sections.
Introduction
- I think it would be helpful to provide some more information about
1) the relevance of all 4 PDK isoforms in the studied tissues (nervous system, muscle) - is one of them predominantly expressed in human muscle and nervous tissue?
2) the affinity/ability to phosphorylate PDH of PDK1 in comparison to the other isoforms.
3) the different phosphorylation sides of PDH and their relevane for PDH activity.
M&M
Point 2.3
- This section is far too short. Please provide detailed information about cell culture conditions and especially about the measurement of PDH activity.
Point 2.4.6
- Please explain why a dosage of 1mM was used. Previous studies? Literature? Have you performed dose-dependency/toxicity experiments?
Point 2.6
- This is far too short. Please provide detailed information about whether and which test was performed to test for normality, which is a must for using ANOVA. In addition, please provide information about the used post-hoc test for the actual comparisons between the different experimental groups. Which version of Prism was used?
Results+Discussion
Point 3.1
- As you evaluated muscle integrity in your zebrafish model, some information about the general muscle histopathology in the patient would be interesting.
- I think the developed precocious puberty is apart from the much earlier developed neurological phenotype a major finding, especially since apparently she had several septic episodes. Could you provide more information, if available, about the endocrine status (Glucocorticoid vs. sex hormones?), any changes in immune cells?
Point 3.3
- Please refer to my comment for the respective M&M part. If you do not show any primary data for this assay, please explain more the basis of the described reference range (control cells that were processed in technical replicates in the same experiment as the patient cells? reference range of a standard diagnostic kit, provided by the developer?)
Point 3.4
- You write that the zebrafish orthologue has over 77% similarity with the human protein on amino acid level. Please state whether the respective domain/region in which the patient variant was found is identical between humans and zebrafish and in addition whether there are any known differences in zebrafish PDK1 function compared to the human protein. This leads to the question whether your human PDK1-WT has any different effects than the normal zebrafish pdk1?
- "Data not shown" for the central claim that there is no difference in the overall growth of the embryos is not acceptable. If there are no differences in growth/size during development, then show the data that there are no differences.
- In general an appropriate control is missing: How are the phosphorylated PDH levels changed in the different conditions? Showing that indeed PDH phosphorylation is reduced in the pdk1-MO as well as the pdk1-MO + PDK1G380D group and is restored by PDK1-WT overexpression is a must for all other conclusions drawn from the experiments performed in this study. In addition a western blot should be performed for protein levels of PDK1 in the used experimental groups.
Point 3.5
Fig.2A:
-To what refers the "n=40-45"? Number of embryos or counted cells in one embryo - please specify, also for all other figure panels? How were signals from different cells separated, i.e. have you labelled cells in general, so you know which signal comes from which cell?
I also do not agree that based on different numbers of Mitotracker+ cells you can conclude that there are differences in mitochondria numbers, as you state in the text. As to my knowledge mitotracker fluorescence intensity per cell is a surrogate parameter for mitochondrial mass, i.e. the number of mitochondria per cell, I think this would be the appropriate analysis to perform. Same for Fig. 2E.
- As stated for Point 2.6., please include informations which statistical comparisons were done with which test. If one comparison is not significant, please include this in the figure.
Fig. 2B:
- Why do the pdk1-MO embryos show no difference? If this phenotype is due to differences in energy production, as you state, ther should be a change in muscle activity. Please discuss. Please state to what the n=108-120 refer - embryos in each group?
- In the text you state that the reduced number of coils is due to a defective energy production. This statement is not backed by your results, you only investigated mitochondrial numbers/integrity and muscle integrity. Unless you do not measure ATP levels you can not state that you have a defective energy production. In addition there might be peripheral nervous system differences in the innervation of the muscles you have not ruled out. If you want to keep this statement, then you should measure at least ATP levels in your experimental cohorts.
Fig. 2C:
- Why do the PDK1-WT embryos have lower (even lower than the pdk1-MO only embryos?) number of connections? Please explain and discuss.
Fig. 2D:
- From n=2 you can not draw any conclusion about "that the increase in apoptosis in the experimental groups compared to the control did not explain the phenotype". Please repeat this experiment at least one more time and provide evidence for this statement. In addition, your PDK1-WT group shows the highest apoptosis marker levels - why? Suggests toxicity of this construct.
Actually you have in all 3 interventions a lower number of connections (Fig.2C) and in all 3 interventions increases in Caspase-3 levels. This looks to me like apoptosis indeed can explain the phenotype. However, to be ableto actually state this, you need to include more than 2 biological replicates for your Caspase 3 WB.
- Have you measured only cleaved Caspase 3 levels or total levels? Please specify, also in the figure.
Fig. 2E:
- Please provide an appropriate western blot control that the used inhibitor actually reduces PDH-phosphorylation - samples from the same experimental groups as used in Fig.2E would be helpful. However, at least a DMSO vs. PS10 comparison would be necessary.
Line 358
You have not shown that in zebrafish overexpressing PDK1-G380D PDH phosphorylation is changed, therefore you can not claim this - until you have done the appropriate experiment, as stated above.
Line 366
You state "Even though we did not detect a significant increase in apoptosis" - yes, because you could not perform a statistical test with n=2. Please do - at least - one more replicate as stated above.
Line 377-379
As stated above, you can not claim that the phenotype is due to "a defect in energy production" as long as you have not measured the energy production capability - ATP levels.
Furthermore, please discuss whether the reduction in spontaneous muscle activity/coiling can maybe be explained by the central nervous system changes in terms of reduced of number of connections?
Author Response
Introduction
- I think it would be helpful to provide some more information about
1) the relevance of all 4 PDK isoforms in the studied tissues (nervous system, muscle) - is one of them predominantly expressed in human muscle and nervous tissue?
2) the affinity/ability to phosphorylate PDH of PDK1 in comparison to the other isoforms.
3) the different phosphorylation sides of PDH and their relevane for PDH activity.
We thank the reviewer to bring these points up. For those we have information, we have added it to the manuscript. We couldn’t, however, find many publications regarding the point 3 in the literature. We know from northern blot experiments which PDK are expressed in which tissues and their in vitro kinase activity. It is also known which Serine residues are phosphorylated by each PDK, and the rate of the reaction as well as the de-phosphorylation by PDPs. However, it is not known how it affects the activity of PDC itself (in converting pyruvate in acetyl-CoA).
This information was added to the introduction as follows:
Lines 57-64: “While in the skeletal muscle the most abundant isoforms are PDK2 and PDK4, in the brain northern blot analysis showed that PDK2 is the most abundant isoform, followed by PDK1. Conversely, in vitro kinetic analysis of PDKs activity showed that the isoforms with the highest kinase activity are either PDK1 or PDK3, depending on the publication, followed by PDK4 and lastly PDK2 [9, 10]. Interestingly, the rate of reactivation of PDC by PDPs is dependent on the PDK that phosphorylated PDH, with the fastest dephosphorylation occurring following PDK2 activity, then PDK3, PDK4, and finally PDK1 [10].
M&M
Point 2.3
- This section is far too short. Please provide detailed information about cell culture conditions and especially about the measurement of PDH activity.
We have added the following information to section 2.3:
Lines 122-124: “PDH activity of unstimulated cells was measured in the lab of Gary Brown, Medical Genetics Laboratories, Oxford University, using a commercially available analysis [38].”
Point 2.4.6
- Please explain why a dosage of 1mM was used. Previous studies? Literature? Have you performed dose-dependency/toxicity experiments?
We used PS10 at a concentration of 1mM following a dose/toxicity experiment. This was added to the text as follows:
Lines 177: “…, determined by a previous dose/toxicity experiment.”
Point 2.6
- This is far too short. Please provide detailed information about whether and which test was performed to test for normality, which is a must for using ANOVA. In addition, please provide information about the used post-hoc test for the actual comparisons between the different experimental groups. Which version of Prism was used?
We apologize for the extremely short section. We have added more information in the section, as follows:
Line 214-220: “Experiments were performed in triplicate and quantifications were statistically compared using GraphPad Prism version 9. For data that did not fit the normal distribution of values, a Ln(value) function was applied, and normality was tested and confirmed. Following one-way ANOVA, multiple comparison tests were performed to identify which groups were significantly different. Statistical significance was considered when p<0.05. Non-significant comparisons were not plotted in the graphs. For each comparison, n represents the number of embryos analyzed per group.”
Results+Discussion
Point 3.1
- As you evaluated muscle integrity in your zebrafish model, some information about the general muscle histopathology in the patient would be interesting.
The patient muscle biopsy showed normal muscle tissue. However, during the autopsy the skeletal muscle under high magnification in electron microscopy they observe electron dense inclusions (1-2 microns) of unknown origin. This has been clarified in the patient description.
Lines 254-258: “Muscle biopsy indicated normal ATP-production and normal function of the complexes in the respiratory chain. Muscle histology was normal in clinical routine histological stainings, however, during the autopsy, skeletal muscle investigations by electron microscopy showed the presence of electron dense in-clusions (1-2 m in size) of unknown origin.”
- I think the developed precocious puberty is apart from the much earlier developed neurological phenotype a major finding, especially since apparently she had several septic episodes. Could you provide more information, if available, about the endocrine status (Glucocorticoid vs. sex hormones?), any changes in immune cells?
We agree with the reviewer that this is of high interest, unfortunately cortisol was only measured once and it was normal (577 nmol/L, ref 170-500). This was however measured during a septic episode where one expects a high value and could potentially indicate a slight reduced cortisol response. However, this is very speculative and since no cortisol curve was performed this result was not included in the current paper.
Of note, puberty did not progress, she only started having signs. Regarding the endocrine status, here are the values: TSH 1,52; T4 21; T3 6,4; FSH 1,91 IE/L (reference 0,1-2); LH <0,1 IE/L (ref < 0,3), Estradiol <40 pmol/L (reference <100), Testosterone 1,19 nmol/l, DHEA 1,25 mmol/L (reference <0,5); 17 alfa OH 2,1 nmol/L (reference <2,4); ACTH 33 ng/L.
Immune parameters were only measured during septic episodes, and she showed normal responses. We therefore suspect that her immune system was generally unaffected.
Point 3.3
- Please refer to my comment for the respective M&M part. If you do not show any primary data for this assay, please explain more the basis of the described reference range (control cells that were processed in technical replicates in the same experiment as the patient cells? reference range of a standard diagnostic kit, provided by the developer?)
We have added the information about the assay to the section 2.3 as mentioned above and the normal range was written in the results section.
Point 3.4
- You write that the zebrafish orthologue has over 77% similarity with the human protein on amino acid level. Please state whether the respective domain/region in which the patient variant was found is identical between humans and zebrafish and in addition whether there are any known differences in zebrafish PDK1 function compared to the human protein. This leads to the question whether your human PDK1-WT has any different effects than the normal zebrafish pdk1?
Thank you for the very pertinent comment.
When the amino acid sequences are compared, the amino acid affected by the mutation (380) and those in the vicinity are conserved between human and zebrafish. In fact, most of the differences are found within the first 50 amino acids (N-terminal) and the kinase domain spanning amino acids 163-393 is very similar.
There is very little research in the comparison of PDK function between species, especially to what concern the zebrafish. However, given the level of conservation and the vast number of publications showing that orthologues function is conserved, we suggest that PDK function is also conserved. Our confidence in this is further supported by the fact that there is only one Pdk1 in zebrafish. Following an event of genome duplication, more than one orthologue is often found in the zebrafish genome. When this happens, the functional relevance of both orthologues needs to be tested. This is not the case for PDK1. We therefore assumed that wild type human PDK1 would have a similar effect in zebrafish and our results do not suggest otherwise.
We have updated the figure 1 showing the amino acid conservation and added the following sentences in the figure legend and main text:
Lines 298-299: “(B) Partial amino acid sequence alignment including the amino acid affected by the variant shows conservation between human and zebrafish.”
Lines 310-312: “… and high conservation surrounding the amino acid affected by the variant (Figure 1B). The similarity in the protein sequence suggests that the function is also conserved between species.”
- "Data not shown" for the central claim that there is no difference in the overall growth of the embryos is not acceptable. If there are no differences in growth/size during development, then show the data that there are no differences.
We understand the comment and added the images and quantifications to the manuscript, compiled in a new figure 2 (the following figures were renumbered accordingly).
As we believe the reviewer can appreciate, there are no visible abnormalities in the larvae of each group (embryos shown for the figure are representative of the mean larva length measured in each group). Nevertheless, non-parametric comparison of the quantifications showed a significant difference in the total length of the larvae between most groups. We consider the statistical difference, which corresponds to a maximum difference of 5% in the body length (when control and pdk1-MO + PDK1wt are compared), combined with a lack of visible developmental phenotypes, not to have a significant biological meaning.
We added a new figure 2 and added the following sentences in the main text:
Lines 319-322: “Even though the larvae in the experimental groups were statistically smaller than those in the control group, the larger difference, between control and pdk1-MO + PDK1wt , is 161 mm, which corresponds to a 5% difference in body length. This finding, combined with a normal appearance of the larvae, suggests that the expression of mutant PDK1 does not have a major impact in the overall embryo growth during early development.”
- In general an appropriate control is missing: How are the phosphorylated PDH levels changed in the different conditions? Showing that indeed PDH phosphorylation is reduced in the pdk1-MO as well as the pdk1-MO + PDK1G380D group and is restored by PDK1-WT overexpression is a must for all other conclusions drawn from the experiments performed in this study. In addition a western blot should be performed for protein levels of PDK1 in the used experimental groups.
The comment of the reviewer is valid, and we wanted to do such experiment. However, one of the disadvantages of working with animal models, especially zebrafish or other non-mammalian models, is that the availability of working antibodies is much smaller and that is the case for the antibodies in this study. We tried using the antibodies against PDHA1 and PDH-E1alpha pSer232 in our zebrafish protein extracts but unsuccessfully. We therefore could not include those experiments in the manuscript. To what concerns PDK1, we did not try it, even though we understand that, if working, would have been beneficial to the project.
Point 3.5
Fig.2A:
-To what refers the "n=40-45"? Number of embryos or counted cells in one embryo - please specify, also for all other figure panels? How were signals from different cells separated, i.e. have you labelled cells in general, so you know which signal comes from which cell?
I also do not agree that based on different numbers of Mitotracker+ cells you can conclude that there are differences in mitochondria numbers, as you state in the text. As to my knowledge mitotracker fluorescence intensity per cell is a surrogate parameter for mitochondrial mass, i.e. the number of mitochondria per cell, I think this would be the appropriate analysis to perform. Same for Fig. 2E.
- As stated for Point 2.6., please include informations which statistical comparisons were done with which test. If one comparison is not significant, please include this in the figure.
The reviewer is correct on the fact that we did not count mitochondria, and this was a mistake that was corrected throughout the manuscript. We in fact quantified the number of cells containing labelled mitochondria.
We apologize that the details on the quantifications were not clear. The numbers described in the figure legends are number of embryos analyzed per group. This is consistent throughout the manuscript. In the example given by the reviewer, in this experiment each experimental group had between 40 and 45 embryos in which cells were quantified. Then on average, in each control embryo we counted approximately 32 labelled cells, and so on. We added this information in the section 2.6 Statistical analysis.
Lines 219-220: “Non-significant comparisons were not plotted in the graphs. For each comparison, n represents the number of embryos analyzed per group.”
We have precisely added a higher magnification image in the figure so that the readers could see how easy it is to count the cells. We understand that the figures showing the entire tail of the embryos do not make that very clear. Here, we counted labelled cells, which means cells that had mitotracker in their cytoplasm. We agree that our working was not accurate and changed it throughout the text to “cells with labelled mitochondria”.
Generally, unless the type of experiment warrants showing that differences are not significant (e.g. following treatment), we avoid plotting them in the graph so it does not become too busy. This explanation was written in the section 2.6.
Fig. 2B:
- Why do the pdk1-MO embryos show no difference? If this phenotype is due to differences in energy production, as you state, ther should be a change in muscle activity. Please discuss. Please state to what the n=108-120 refer - embryos in each group?
Since the morpholino used in this project had already been tested and published (Sips et al., 2018), and the results demonstrated in Figure 1A showed an effect when compared to the control embryos, we did not further study the differences between control and pdk1-MO, for example, to what concerns the coiling experiments. The morpholino was used to reduce the amount of endogenous Pdk1 to better recapitulate the heterozygous expression of mutant PDK1 in the patient. However, we could expect that injecting a higher concentration of morpholino would result in a difference in the other essays as well. The main focus of these experiments was to perform the rescue with the wild type or the mutant human mRNA, we therefore were more interested in the differences between the pdk1-MO + PDK1wt and pdk1-MO + PDK1G380D.
The “n” were clarified in all figure legends.
- In the text you state that the reduced number of coils is due to a defective energy production. This statement is not backed by your results, you only investigated mitochondrial numbers/integrity and muscle integrity. Unless you do not measure ATP levels you can not state that you have a defective energy production. In addition there might be peripheral nervous system differences in the innervation of the muscles you have not ruled out. If you want to keep this statement, then you should measure at least ATP levels in your experimental cohorts.
We have changed our statements to reflect the fact that we did not measure ATP production from these embryos.
Lines 510-513: “… we suggest that there may be a defect in energy conversion, likely from a decrease in functional mitochondria. As glycolysis results in less energy conversion than the mito-chondria-driven OXPHOS [63] which, we suggest, accounts for the reduction in spon-taneous skeletal muscle contraction.”
Fig. 2C:
- Why do the PDK1-WT embryos have lower (even lower than the pdk1-MO only embryos?) number of connections? Please explain and discuss.
This is a very interesting point, and we refer to the comment above. The morpholino was used to reduce the amount of endogenous protein, to better model the patient. It is not surprising that any approach to evaluate the role of a variant is not free of some side-effects. Here, we suspect that the overexpression of a gene in every cell of the body may result in some toxicity. In this study we were focused on evaluating the effects of mutant PDK1 in development, therefore we were mostly interested in comparing the wide type and mutant mRNA.
We think nevertheless that this finding would warrant further investigation. However, in order to study such complex anatomy a stable line is needed which is beyond the scope of this publication.
Fig. 2D:
- From n=2 you can not draw any conclusion about "that the increase in apoptosis in the experimental groups compared to the control did not explain the phenotype". Please repeat this experiment at least one more time and provide evidence for this statement. In addition, your PDK1-WT group shows the highest apoptosis marker levels - why? Suggests toxicity of this construct.
Line 366
You state "Even though we did not detect a significant increase in apoptosis" - yes, because you could not perform a statistical test with n=2. Please do - at least - one more replicate as stated above.
We agree with the comment, and we have rephrased the sentence to better reflect our hypothesis. We suggest that the increased apoptosis does not fully explain the phenotypes, however we cannot exclude that it may have a partial role.
Actually you have in all 3 interventions a lower number of connections (Fig.2C) and in all 3 interventions increases in Caspase-3 levels. This looks to me like apoptosis indeed can explain the phenotype. However, to be ableto actually state this, you need to include more than 2 biological replicates for your Caspase 3 WB.
- Have you measured only cleaved Caspase 3 levels or total levels? Please specify, also in the figure.
We have quantified the cleaved caspase3 and added that information to the text.
Line 407: “(quantification of cleaved Caspase-3…)
Fig. 2E:
- Please provide an appropriate western blot control that the used inhibitor actually reduces PDH-phosphorylation - samples from the same experimental groups as used in Fig.2E would be helpful. However, at least a DMSO vs. PS10 comparison would be necessary.
Unfortunately, the antibodies do not work in zebrafish samples, so we could not do this experiment.
Line 358
You have not shown that in zebrafish overexpressing PDK1-G380D PDH phosphorylation is changed, therefore you can not claim this - until you have done the appropriate experiment, as stated above.
We would have liked to add WB quantifications to this experiment but unfortunately the antibodies didn’t work in zebrafish protein extracts, therefore we were unable to do so.
Line 377-379
As stated above, you can not claim that the phenotype is due to "a defect in energy production" as long as you have not measured the energy production capability - ATP levels.
Furthermore, please discuss whether the reduction in spontaneous muscle activity/coiling can maybe be explained by the central nervous system changes in terms of reduced of number of connections?
We appreciate the reviewer’s observation and that measuring the energy conversion capacity of these embryos was something we wanted to do but were not able to.
In this case, we used the coiling because it is independent of neuronal connections originated in the brain. They were also unstimulated, in contrary to, for example, the touch evoked response assay, which would involve a response mediated by the neurons.
Reviewer 3 Report
Review:
A missense variant in PDK1 associated with severe neurodevelopmental delay and epilepsy.
This is quite an interesting study in relating the human disease to inducing a similar genetic disorder in zebrafish. To investigate the behavior and developmental factors in the zebrafish turned out quite intriguing . The effects of the dysfunction in PDK1 is well characterized in this report.
All aspects of this report are very clear. As far as I can tell no modifications are needed.
Author Response
The authors thank the reviewer for the positive comments.
Round 2
Reviewer 2 Report
Introduction
Fine, apart from the sentence:
"lead to a reduced number of cells with detectable membrane potential..."
- I think the word mitochondrial is missing,
so the sentence should read
"lead to a reduced number of cells with detectable mitochondrial membrane potential..."
M&M
Fine, apart from:
2.3 Cell culture and oxygen consumption analysis 108
A skin biopsy was obtained from which fibroblasts were grown. PDH activity of unstim-109 ulated cells was measured in the lab of Gary Brown, Medical Genetics Laboratories, Ox-110 ford University, using a commercially available analysis [38].
--> Please insert the name of the company and test with catalogue number - only citing Dr. Browns paper from 1986 is not appropriate.
2.6 Statistical analysis
Ln transformation instead of using simply a non-parametric test - okay, acceptable.
However, you still do not declare with which test you decided whether your data were normally or not normally distributed.
--> Insert which normality test was used!!
"Following one-way ANOVA, multiple comparison tests were performed to identify which groups were significantly different."
--> Insert which multiple comparison test was used!!
Results
Points 3.1 to 3.3
Thank you for the additional informations, no further comments.
Point 3.4
I agree that there is no major effect on overall growth. However you write here that you've conducted a "non-parametric" analysis.
This is not covered by the according M&M section where you stated that you Ln transformed non-normally distributed data and conducted then
an ANOVA.
(1) What non-parametric test was used?
(2) What post-hoc test was used?
(3) Why not a Ln transformation as stated in the statistics part?
Point 3.5
No further comments.
Discussion:
I believe you that apparently none of the antibodies for PDH/P-PDH/PDK1 is working in zebrafish.
However, it is clear that therefore you do not have any kind of control that basically any experiment
you've performed is appropriately working through regulation of PDH phosphorylation. If one would be strict,
one would need to state that you can not conclude from the experiments you performed that the results have anything
to do with differential PDH phosphorylation.
On the other hand I agree that it would be rather unlikely that the phenotypes observed are mediated by another mechanism
apart from the well known PDH-modulating effects of PDK1. Nevertheless, I think it is severely misleading to completely ignore this
major issue that you have not confirmed on protein level that your constructs actually lead to the effect you assume, therefore please - at least -
mention and discuss this issue as a limitation of the animal experiments you have performed.
Overall comment:
It is in my eyes a must that the authors clarify the remaining major issues, especially the inconsistent statistics and the discussion of the major limitation regarding the control experiments that could not be performed due to not-working antibodies.
If the authors are able to do this, the manuscript can be accepted for publication.
Author Response
Introduction
Fine, apart from the sentence:
"lead to a reduced number of cells with detectable membrane potential..."
- I think the word mitochondrial is missing,
so the sentence should read
"lead to a reduced number of cells with detectable mitochondrial membrane potential..."
This issue has been fixed in the text (line 84).
M&M
Fine, apart from:
2.3 Cell culture and oxygen consumption analysis 108
A skin biopsy was obtained from which fibroblasts were grown. PDH activity of unstim-109 ulated cells was measured in the lab of Gary Brown, Medical Genetics Laboratories, Ox-110 ford University, using a commercially available analysis [38].
Please insert the name of the company and test with catalogue number - only citing Dr. Browns paper from 1986 is not appropriate.
We are not able to provide a catalogue number as this test was developed in house.
Below are the changes in the text we have made:
Line 106-112: “ A skin biopsy was obtained from which fibroblasts were grown. PDH activity of unstimulated cells was measured in the lab of Gary Brown, Oxford Regional Genetics Laboratories, Oxford University Hospital NHS Foundation Trust,, using a locally available analysis, available upon request [38]. After maximal activation of PDC by dichloroacetate, cells are incubated with [1-14C] pyruvate and the resulting 14CO2 generated is quantified in a liquid scintilation counter. The normal range for this assay is 0.6-0.9 nmol/mg protein/minute.”
2.6 Statistical analysis
Ln transformation instead of using simply a non-parametric test - okay, acceptable.
However, you still do not declare with which test you decided whether your data were normally or not normally distributed.
At the time we ran the non-parametric tests too and they gave us consistent results. For the manuscript, we decided to use parametric tests as often as possible, which happened for all the tests expect the one in Figure 2, comparing the body lengths.
The normal distribution of values was tested using the D’Agostino & Pearsons test. When data did not follow a normal distribution of values, we use the ANOVA non-parametric Kruskal-Wallis test and the Dunn’s multiple comparisons test. For the comparisons where the parametric test could be performed, we used the one-way ANOVA and the Turkey’s multiple comparisons test.
--> Insert which normality test was used!!
"Following one-way ANOVA, multiple comparison tests were performed to identify which groups were significantly different."
--> Insert which multiple comparison test was used!!
The material and methods section was changed as follows:
Line 203-210: “For data that did not fit the normal distribution of values using the D’Agostino and Pearsons test, a Ln(value) function was applied, and normality was re-tested and confirmed. Statistical analysis of data that did not follow the normal distribution of values was performed using the Kruskal-Wallis test, followed by the Dunn’s multiple comparison test (statistical analysis shown on Figure 2). Conversely, one-way ANOVA was performed when normality was verified, and the Turkey’s multiple comparisons test was performed to identify which groups were significantly different.”
Results
Points 3.1 to 3.3
Thank you for the additional informations, no further comments.
Point 3.4
I agree that there is no major effect on overall growth. However you write here that you've conducted a "non-parametric" analysis.
This is not covered by the according M&M section where you stated that you Ln transformed non-normally distributed data and conducted then an ANOVA.
(1) What non-parametric test was used?
(2) What post-hoc test was used?
(3) Why not a Ln transformation as stated in the statistics part?
We have added the information about the statistical analysis in the M&Ms section. Here, we tried several ways to transform the data but the morpholino group did not fit the normal distribution, therefore, we decided to run the non-parametric Kruskal-Wallis test, followed by the Dunn’s multiple comparisons test.
Point 3.5
No further comments.
Discussion:
I believe you that apparently none of the antibodies for PDH/P-PDH/PDK1 is working in zebrafish.
However, it is clear that therefore you do not have any kind of control that basically any experiment you've performed is appropriately working through regulation of PDH phosphorylation. If one would be strict, one would need to state that you can not conclude from the experiments you performed that the results have anything to do with differential PDH phosphorylation.
On the other hand I agree that it would be rather unlikely that the phenotypes observed are mediated by another mechanism apart from the well known PDH-modulating effects of PDK1. Nevertheless, I think it is severely misleading to completely ignore this major issue that you have not confirmed on protein level that your constructs actually lead to the effect you assume, therefore please - at least - mention and discuss this issue as a limitation of the animal experiments you have performed.
We agree with the reviewer, and we have added the following sentences to the discussion:
Line 438-446: ”Altogether, the compiled results suggest that the clinical phenotype of the patient as well as the phenotypes observed in the zebrafish are mediated by abnormalities in the PDC activity. However, since the available antibodies to detect the phosphorylated PDH do not work in zebrafish, we were not able to confirm this mechanism. This limitation needs to be taken into account. Another major issue is the variability of phenotypes observed in both morphant and PDK1wt overexpressing embryos. It is unlikely that the transient approach used in this study will be able to address such questions. Further experiments using patient cells and stable mutant animal models would be better suited to fully explain the mechanisms of PDK1-associated disorders.”
Overall comment:
It is in my eyes a must that the authors clarify the remaining major issues, especially the inconsistent statistics and the discussion of the major limitation regarding the control experiments that could not be performed due to not-working antibodies.
If the authors are able to do this, the manuscript can be accepted for publication.
We hope that the reviewer is satisfied with the changes.